# Crack damage stress in fully saturated rocks: A new detection procedure

Sandra Schumacher[1] and Werner Gräsle[1]

[1]Federal Institute for Geosciences and Natural Resources (BGR), Hanover, Germany

**Correspondence:** Sandra Schumacher (sandra.schumacher@bgr.de)

**Abstract.** The crack damage stress also known as onset of dilantacy gives the long-term strength of a rock as it describes the transition from the stable to the unstable crack growth phase under loading. As such is it of significant interest e.g. for long-term safety analyses of radioactive waste repositories. These long-term safety analyses are based on numerical models and thus require the incorporation of a constitutive equation for the crack damage stress. However, such a constitutive equation is still
missing as a precise determination of the crack damage stress is required to establish parameter dependencies. In this study, we propose a new procedure to determine the crack damage stress which combines an innovative measurement technique using pore pressure diffusion with the well known technique of finding the pore pressure maximum. The new technique monitors the true axial strain as indicator for the crack damage stress during a pore pressure diffusion test. In addition to the crack damage stress, this new true axial strain method simultaneously yields pore pressure diffusion coefficients, thereby maximising the
information gain. The true axial strain method was developed using a Bunter Sandstone sample, but it can be applied to other types of rocks, which is demonstrated on a multi-cycle, long-term experiment of one sample of Passwang Marl.

## 1 Introduction

Understanding the hydro-mechanical behaviour of rocks and soils under stress is of significant interest for many applications such as oil and gas extraction (e.g. David and Ravalec-Dupin, 2007; Zoback, 2007; Ahmed, 2018), nuclear waste disposal (e.g.
Belmokhtar et al., 2017; Minardi et al., 2020; Crisci et al., 2024), $CO_2$ storage (e.g. Zhou et al., 2010; Williams et al., 2014; Allen et al., 2020), geothermal energy production (e.g. Medici et al., 2023; Zhang et al., 2023a), sediment transport (e.g. van Damme, 2021) or levee construction (e.g. Girardi et al., 2023). For applications which require the soil or rock mass to provide an effective seal against fluid transport, a more precise knowledge of the crack damage stress ($\sigma_{CD}$) is of vital importance as new fluid pathways form in the dilatant regime (e.g European Commission et al., 1999; Naumann et al., 2007; Heap and
Wadsworth, 2016; Zhao et al., 2021). Moreover, the crack damage stress defines the long-term strength of a rock under loading as it marks the transition from the stable into the unstable crack growth regime (e.g Martin and Chandler, 1994; Zhang et al., 2023b).

Much work has already been done on the theoretical description of the processes leading to cracks and the processes occurring within the material during cracking. Most of these works are based on general ideas about cracking in solid materials but
the fundamental concepts described therein can also be applied to geomaterials such as rocks. The ideas first formulated by

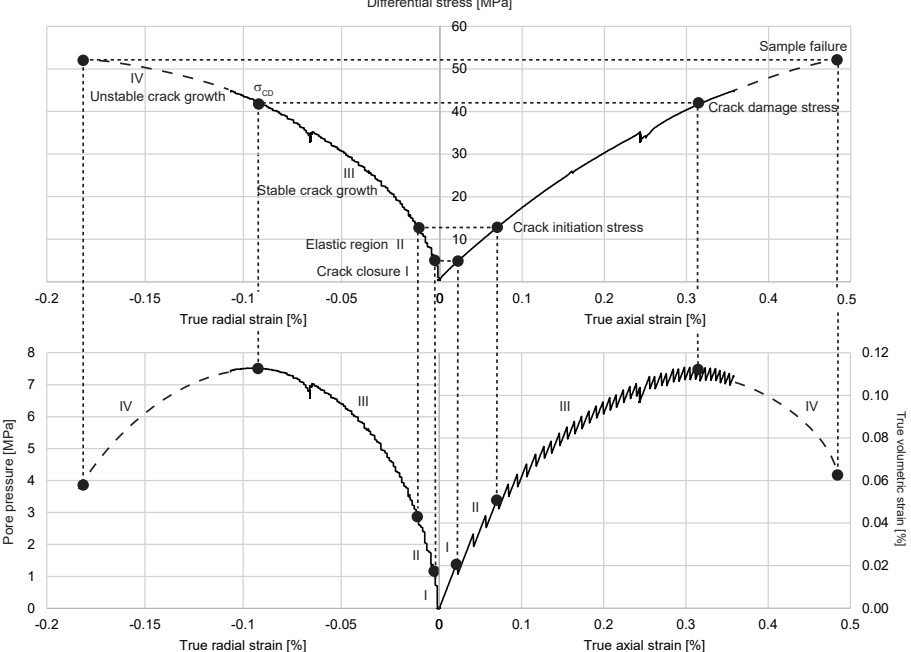

**Figure 1.** Sketch of the stress-strain diagram partially using measurement data from the Bunter Sandstone sample, which are indicated by the continuous lines. Also the pore pressure evolution and the evolution of the true volumetric strain during are shown. Dashed lines indicate extrapolated evolution curves.

Griffith and Taylor (1921) concerning cracks are e.g. still widely used in rock mechanics (e.g. Jaeger and Cook, 1979; Charlez, 1991; Fossen, 2010). Over the years, many studies have explored and expanded the theoretical background to crack formation (e.g. Landau and Lifshitz, 1976; Cherepanov, 1979; Atluri, 1986; Parton, 1992; Aleinikov et al., 2023).

During loading, two processes compete with each other. On the one hand, existing pores and cracks close, resulting in a
compaction of the sample. On the other hand, the increasing load leads to the formation of new cracks, which causes the sample to expand (Heap and Wadsworth, 2016). The behaviour of the rock due to these competing processes can be divided into four different phases (e.g. Martin and Chandler, 1994; Martin et al., 2001; Cai et al., 2004; Xue et al., 2014; Zhang et al., 2023b): In the first stage, the closure of pre-existing cracks and pore space occurs (Fig. 1) and the bulk sample volume decreases because of this. In the second stage, the rock deforms elastically, while in the third stage new fractures form within
the rock while the closure of previously existing fractures and pores continues. Up until now, the bulk sample volume still decreases as the closure of pre-existing cracks outweighs the formation of new fractures. In the fourth stage, however, unstable crack growth sets in as cracks start to coalesce, ultimately resulting in the failure of the sample. This transition from stable to unstable crack growth is called the crack damage stress and the formation of new cracks will lead to an increase in bulk sample volume despite the continuing increase in load (Fig. 1).

This change from bulk sample volume decrease to increase under increasing differential stress is often used to define the crack damage stress (e.g. Martin, 1997; Palchik and Hatzor, 2002; Corkum, 2020; Taheri et al., 2020; Wen et al., 2023; Zhang et al., 2023b; Li et al., 2023; Xiao et al., 2023), which is identical to the onset of dilatancy (Alkan et al., 2007; Wu et al., 2018; Zhang, 2018) or the point of critical energy release (Bieniawski, 1967). Some authors (Chang and Lee, 2004; Taheri et al., 2020) define the crack damage stress as the point in the volumetric stiffness curve where stiffness values changed from positive to negative or as the point where cumulative acoustic emission curves increase abruptly. Under true triaxial conditions, also the 0.5 value of the instantaneous Poisson's value $\mu_{13}$, which is the ration of the $\epsilon_3$ and $\epsilon_1$ strain rates, is considered as indicator for the crack damage stress (Kong et al., 2018). The crack damage stress can also be detected by identifying the axial stress at which the axial stress-volumetric strain curve reverses; a method which is often employed for unsaturated rocks under uniaxial compression (e.g. Martin, 1997; Palchik and Hatzor, 2002; Xue et al., 2014; Taheri et al., 2020) or true triaxial loading conditions (Gao et al., 2020; Zhang et al., 2023b). Especially for rocks in which no reversal of the axial stress-volumetric strain can be observed prior to failure, the stress threshold determination method using axial and lateral crack strains is proposed to determine the crack damage stress (Mo et al., 2024). Moreover, the crack damage stress can be identified by analysing acoustic emissions (Chang and Lee, 2004; Wu et al., 2021; Abbas et al., 2023).

In fully saturated rocks, tests under confined and undrained conditions (CU tests) can be used to determine the crack damage stress without measurements of the radial strain. In CU tests, the pore pressure is expected to increase with increasing load as the sample compacts. When the crack damage stress is reached, the sample volume starts to increase, which in turn leads to a drop in pore pressure (Fig. 1). Thus, in CU tests, the pore pressure maximum can act as an indicator for the reverse in the axial stress-volumetric strain curve and thus as an indicator for the occurrence of the crack damage stress (Yoshinaka et al., 1997; Brandon et al., 2006).

Although the theoretical background to all of the above-described measurement techniques is well defined, achieving perfect boundary conditions in the laboratory is impossible. Each and every measurement technique is influenced by unwanted effects which can be mitigated but rarely – if ever – completely removed. Such unwanted effects e.g. can be the response of the triaxial testing machine on the measurements such as the storage in the tubing system between sample and pore pressure transducer or fluctuations in the room temperatures which influence the recorded pore pressure. Also small leakages within the system which develop over time are a potential problem which prevents measurements under perfect conditions. Thus, it is advisable to combine different detection methods which suffer from different unwanted influences in order to obtain the most reliable results for the crack damage stress. However, in case of CU tests on fully saturated samples, no immediate complementing method to the determination via the maximum pore pressure during loading exists. This is a problem as such tests – especially when performed on low-permeable rock and over a long period of time – tend to suffer e.g. from pore pressure leakages, temperature variations and drifting of sensors.

The objectives of our work were therefore to:

(i) find a measurement method which does not suffer from small pore pressure leakages like the methods which relies on the maximum pore pressure to detect the crack damage stress

(ii) detect the crack damage stress with the highest possible accuracy even in long-running experiments

(iii) prove that the new method works for all types of fully saturated rocks

We implemented this by testing a new method to measure the crack damage stress using a Bunter Sandstone sample. Combining this new method with a conventional measurement method, we developed a new experimental procedure which allows to validate the results of both methods. Using a long-running triaxial experiment of Passwang Marl, we show that the new procedure cannot only be applied to sandstones but to other types of fully saturated rocks as well even if permeabilities are as 80 low $10^{-20}\mathrm{m}^2$.

## 2   Triaxial testing equipment

The tests were conducted at the laboratory for rock physics of the Federal Institute for Geosciences and Natural Resources (BGR) in Hanover, Germany. Two different triaxial testing apparatuses were used for the two samples investigated in order to save time given that the individual experiments lasted months or years to finish.

### 2.1   Triaxial testing apparatus for the Passwang Marl sample

The high-pressure triaxial testing apparatus employed for the Passwang Marl regulated the cell pressure on demand by means of a loading system with confining pressures up to 40 MPa and axial loads up to 1000 kN. Three linear variable differential transformers (LVDT) with a resolution of 1 $\mu$m measured the axial displacement. The true axial strain ($\epsilon_a$ [%]) was calculated using the arithmetic mean of the change in length these three LVDTs:

$$\epsilon_a = -100 \cdot ln\frac{l_0 - \Delta l}{l_0} \tag{1}$$

where $l_0$ is the initial length of the sample [mm] and $\Delta l$ is the arithmetic mean of the change in length [mm] of the three installed LVDTs. No device to measure the circumferential displacement was installed.

In order to realise hydraulic testing, the back pressure at one front end of the sample (inlet pressure, bottom) could actively be regulated via a pressure pump (VPC 250/200, Wille-Geotechnik), while the back pressure at the other front end (top) could 95 only change passively (outlet pressure) except for the possibility of a complete unloading by opening a valve. Both inlet and outlet pressure were monitored using pressure transducers. From the pressure pump, the pore fluid is transported via a 3 m long tube with an inner diameter of 2 mm to the lower piston of the triaxial testing machine. It then flows through the piston into a porous disc of sintered metal which connects to the face of the sample. The porous disc has a diameter of 100 mm and a thickness of 19 mm and ensures that the pore fluid which is transported through the piston is distributed evenly over the face 100 of the sample. On the other side of the sample, another porous disc is located between sample face and upper piston. The pore fluid can flow through the upper piston into another tube which acts as a small reservoir and can be opened and closed by a valve, e.g. for de-airing the pore fluid system.

The temperature within the triaxial testing apparatus was kept constant at 30 °C $\pm$ 0.01 °C.

Initially, the rate of increase in differential stress was set to result in a true axial strain rate of $1 \cdot 10^{-8} \, \text{s}^{-1}$. However, we realised that instead of aiming at a constant true axial strain rate, a constant increase in differential stress allowed for a smoother regulation of the triaxial testing apparatus, thereby resulting in less noise on the data. Thus, from day 335 of the experiment onwards, instead of using a constant strain rate for loading, we used a constant stress rate of $0.21 \, \text{MPa} \, \text{h}^{-1}$. This differential stress rate relates to an axial strain rate of about $8 \cdot 10^{-9} \, \text{s}^{-1}$.

No circumferential measurement devices such as extensiometer chains were installed as this experiment initially did not aim to demonstrate the validity of the procedure proposed in this study. Instead we used this running experiment of a low permeable rock to prove that the procedure also works for rocks with permeabilities as low as $10^{-20} \text{m}^2$.

## 2.2 Triaxial testing apparatus for the Bunter Sandstone sample

The high-pressure triaxial testing apparatus employed for the Bunter Sandstone regulated the cell pressure on demand by means of a loading system with confining pressures up to 60 MPa and axial loads up to 1500 kN. The true axial strain was measured and calculated as for the Passwang Marl sample. Three circumferential extensiometer chains (MTS 632.92H-03) were installed at 25 %, 50 % and 75 % of sample length as this experiment on the Bunter Sandstone was deliberately set up to provide information on the sample's axial and volumetric deformation in order to substantiate the claims of this study. Using the data from the LVDTs and the extensiometer chains, the true volumetric strain ($\epsilon_v$ [%]) was calculated as:

$$\epsilon_v = 2 \cdot (-100) \cdot ln \frac{r_0 - \Delta r}{r_0} + \epsilon_a \tag{2}$$

where $r_0$ is the initial radius of the sample [mm] and $\Delta r$ the change in radius [mm] calculated from the arithmetic mean of the changes in length of the three extensiometer chains.

The inlet and outlet pressure measurement system was the same as for the Passwang Marl sample, with a different pressure pump (VPC 250/200-15, Wille-Geotechnik) being the only difference.

The strain rate was set to $1 \cdot 10^{-7} \, \text{s}^{-1}$. This is higher than for the Passwang Marl. However, as the Bunter Sandstone is much more permeable, it was possible to use a higher strain rate and still obtain hydraulic equilibrium during the experiment. Thus, in order not to prolong the experiment unnecessarily, a higher strain rate than for the Passwang Marl was chosen.

## 3 Sample consolidation

To demonstrate the validity of the new detection method, samples from two different rock types were used. One sample consists of marl from the lower Passwang Formation, while the second sample consists of Bunter Sandstone.

## 3.1 Passwang Marl

The marl sample used for this study was cored from the lower Passwang Formation at the Mont Terri URL in Switzerland. The sample was cored with an angle of about $45°$ to the bedding (Z-orientation) from borehole BPE-2 (Gygax et al., 2017), with a depth of 55.8 to 56 m. After preparation, the sample had a length of 199.96 mm and a diameter of 99.97 mm.

The depositional enironment of the Passwang Formation was a shallow, mixed siliciclastic and carbonate depositional environment (Burkhalter, 1996), which is backed by the finding of e.g. crinoids and ammonites (Hostettler et al., 2017). The sample used in this study derives from the lower part of the Passwang Formation, which is of late Aalenian age (Hostettler et al., 2017). The Passwang marls show a high variability of the mineralogy. For the sample used in this study, clay-mineral content is expected to be 20 % weight content, the total carbonate 61 %, and the quartz and feldspar 17 %, with K-feldspar weight content reaching 2 %. Total carbonate is mainly made of calcite (50 %), but dolomite/ankerite are present in as much as 11 %, whereas siderite is absent. Pyrite as a major sulphur-bearing phase is present in small amounts of 0.5 % and organic carbon in 1.0 % (Waber and Rufer, 2017).

The porosity of Passwang marls varies between 6.1 and 14.4 % (Waber and Rufer, 2017), while the permeability of this sample is in the range of $10^{-20}$ to $10^{-19} \mathrm{m}^2$.

In order to avoid chemical alterations within the sample, which could lead to changes in the geomechanical properties (Ewy et al., 2008), so-called "Pearson water" with a composition as close as possible to the in-situ conditions was used. (Pearson, 2002)

The sample was consolidated at 20 MPa confining pressure, which was kept constant for almost 27 days in order to close artificial (micro) fractures that formed during sample preparation. During this time, the true axial strain rates decreased from about $3 \cdot 10^{-6} \mathrm{~s}^{-1}$ to less than $1 \cdot 10^{-9} \mathrm{~s}^{-1}$. Towards the end of the consolidation phase, deformation and therefore strain rates were often below the resolution limit of the LVDTs of about $4.7 \cdot 10^{-15} \mathrm{~s}^{-1}$ for measurements taken every minute. After reducing the confining stress to 2 MPa, true axial strain rates rapidly approached values of less than $1 \cdot 10^{-9} \mathrm{s}^{-1}$ again, indicating a consolidated sample.

## 3.2 Bunter Sandstone

The sandstone was derived from the well 4624IG0105 (ID number of the State Authority for Mining, Energy and Geology, Lower Saxony) from a depth of 12.1 to 12.5 m. The well is located in the vicinity of Hannoversch Münden in the very south of Lower Saxony/Germany. It is part of the Solling Formation of the Bunter Sandstone. Sandstones of the Solling Formation in the area of Hannoversch Münden have been identified to be of fluvial origin (Olsen, 1988) and were deposited about 249 Ma ago during the Olenekian (Heunisch et al., 2017; Deutsche Stratigraphische Kommission, 2022). The permeability of Bunter Sandstones has been determined to be in the range of $10^{-17} \mathrm{m}^2$ with porosities of between 9 to 18 % (Breede, 2006).

The main mineral of this sample is quartz with 60 % weight content, followed by orthoclase and muscovite with 11 % each. The sandstone also contains smectite (7 %), plagioclase (3 %), chlorite (3 %), caolinite (2 %), calcite (2 %), haematite (1 %) and traces of rutile.

After preparation, the sample had a length of 199.94 mm and a diameter of 99.95 mm. To saturate the initially completely dry sample of Bunter Sandstone and for the further testing procedure, ordinary tap water was used as chemical alterations within the sample were considered to be irrelevant.

Consolidation of the sample occurred over five days in which the confining pressure was kept at 20 MPa. Over the course of the consolidation phase the strain rate decreased from about $1 \cdot 10^{-5} \mathrm{~s}^{-1}$ to less than $1 \cdot 10^{-10} \mathrm{~s}^{-1}$ where it stabilised.

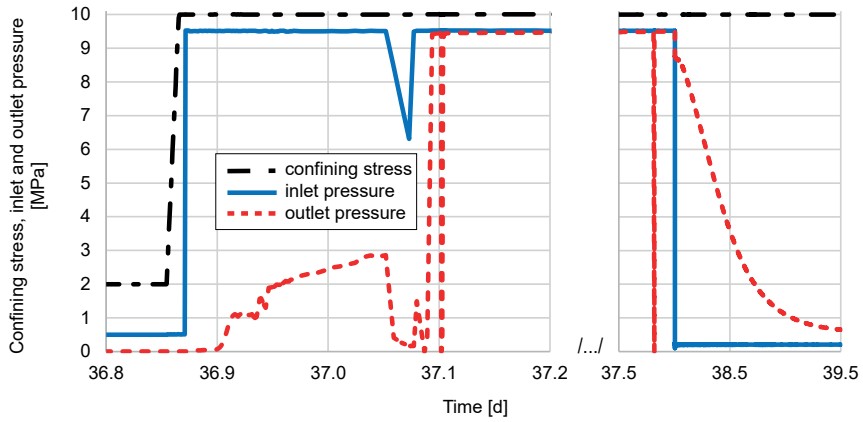

**Figure 2.** The saturation phase for the sample used in this study. Note the different spacing of the x-axis

## 4    Sample Saturation

### 4.1    Passwang Marl

The saturation phase for the this sample was comparatively quick with only three phases in less than 3 days (Fig. 2). In phase 1, with the confining pressure still at 2 MPa, the inlet pressure was set to 0.5 MPa for two days. In phase 2, the confining pressure was increased to 10 MPa and the inlet pressure to 9.5 MPa. While the outlet pressure did not show any reaction to the inlet pressure increase in phase 1, in phase 2 it started to increase after only 5 min. However, initially this increase was too small to be clearly seen in Fig. 2 (from 0.072 bar to 0.095 bar in the first 5 min).

The outlet pressure approached a maximum value of close to 3 MPa after 260 min. Due to technical problems, the inlet pressure decreased to 6.3 MPa over 30 min and increased back to 9.5 MPa within 6 min in phase 3. Less than 30 min later, the outlet pressure nearly reached the same value as the inlet pressure with a difference of less than 0.1 MPa. To verify this result, the valve for the outlet pressure was opened and closed again to observe the outlet pressure increase at a very high data rate of one measurement per five seconds shortly before and after day 37.1. This test showed a rapid increase in outlet pressure from

0 MPa to more than 9.4 MPa in less than 4 min.

A hydraulic short-cut was probably responsible for the observed fast increases in outlet pressure after de-airing shortly before and after day 37.1. This idea is supported by further data. When the inlet pressure was decreased from 9.5 to 2 MPa about a day later, the outlet pressure dropped synchronously for the first 0.8 MPa, as can be expected in case of a hydraulic short-cut, and then settled into a typical pressure diffusion curve. It thus seems that for some part of the saturation process a hydraulic

short-cut was active, which later closed and did not reopen again during the remainder of the experiment.

The saturation of the sample was determined by changing the confining stress under hydrostatic conditions and monitoring the response of the inlet and outlet pressure. The parameter Skempton B, which describes the saturation state of the sample, can be calculated from:

$$B = \frac{\Delta u}{\Delta \sigma_3} \qquad (3)$$

where B is the Skempton B parameter [-], $\Delta u$ is the observed change in inlet or outlet pressure [MPa] and $\Delta \sigma_3$ is the imposed change in confining stress [MPa].

For an effective confining stress of less than 5 MPa, most measurements indicate Skempton B values to be between 0.88 and 0.92. One Skempton B test, however, yielded values of only about 0.75. Yet, these values can be regarded as outliers given the general decrease of Skempton B values with increasing effective confining stress observed for this sample. For an effective confining stress of 10.3 MPa, values decrease to 0.65. Skempton B values derived from the response of the inlet and outlet pressure to the change in confining stress differed slightly across all measurements (see Fig. 3a). The decrease in Skempton B with increasing effective confining stress is in accordance with earlier findings for claystones (Wild et al., 2015; Wild and Amann, 2018; Favero et al., 2018) and other rock types (Mesri et al., 1976; Zimmerman, 1991).

The measurements for the Skempton B values shown in Fig. 3a were performed not only at the beginning but also over the course of the entire experiment. Undrained conditions are required to precisely measure Skempton B. However, these undrained conditions cannot be realised completely as the porous discs at the front ends as well as the linings are also part of the tested reservoir and not only the sample itself. Thus, a certain fluid flow between sample and inlet and outlet reservoir cannot be avoided if the confining stress is changed. As a consequence, the change in measured pore pressure due to a changing confining stress in perfect undrained conditions would be higher than the observed change (Wissa, 1969; Gutierrez et al., 2015). Therefore, the measured Skempton B parameter is likely underestimating real conditions. The Passwang Marl sample was considered to be fully saturated due to the repeatedly measured high value of the Skempton B parameter.

## 4.2  Bunter Sandstone

The confining pressure was kept at 20 MPa for the saturation phase of the Bunter Sandstone. Initially, the inlet pressure was increased to 3 MPa for five days, in which the outlet pressure showed no response. Thus, the inlet pressure was increased to 6 MPa, which was kept constant for 32 days. During this time, the sample was de-aired thirteen times until the outlet pressure response did not change any more. For this, a valve at the outlet pressure side was opened in order to let trapped gas escape to the atmosphere. During this process, pore fluid was present in the fluid lines both sides of the valve to prevent atmospheric gases from entering the system. After this, four tests for Skempton B were performed (Fig. 3b) indicating a fully saturated sample. The tests also confirmed the dependency of the Skempton B parameter on the effective confining stress.

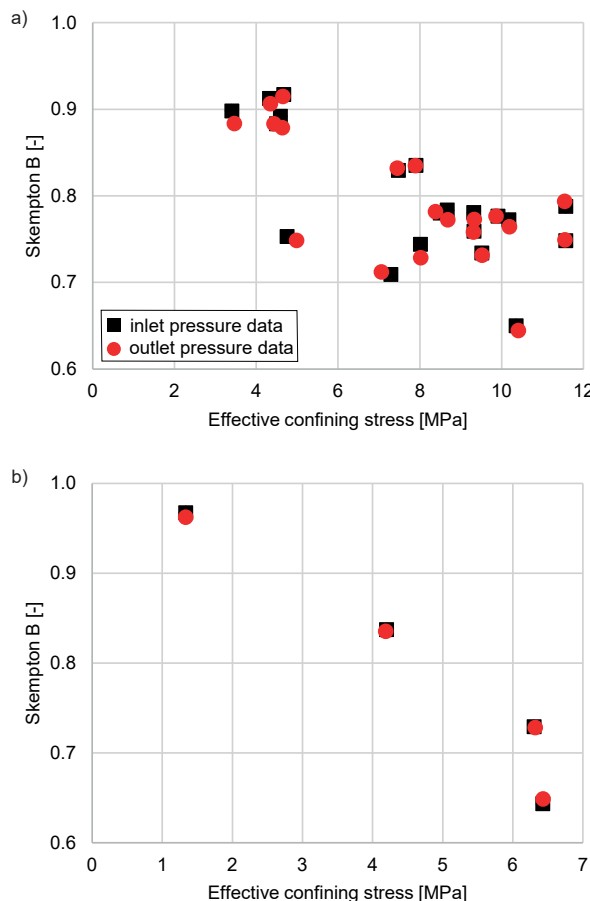

**Figure 3.** Skempton B values for a) the Passwang Marl and b) the Bunter Sandstone sample

## 5   Proposed new procedure

In this study, a new method is presented which allows the determination of the crack damage stress of a fully saturated sample under confined and drained conditions (CD test). This new method is compared with the conventional method used in CU tests. The conventional method is hereafter called maximum pore pressure method (MPP method), while the new method is called true axial strain method (TAS method). The combination of both methods yields the proposed new experimental procedure, which is shown in Fig. 4. It depicts a single cycle of the entire experiment, which consists of repeated cycles at different effective confining stresses. In the cycle shown in Fig. 4, the MPP methods encompasses phases 1 to 3, while the new TAS method is covered by phases 4 and 5 (detailed below).

In Fig. 4, the denoted time refers to the time since the start of the experiment cycle. In all subsequent figures, the denoted time refers to the start of a single phase of the respective experiment cycle.

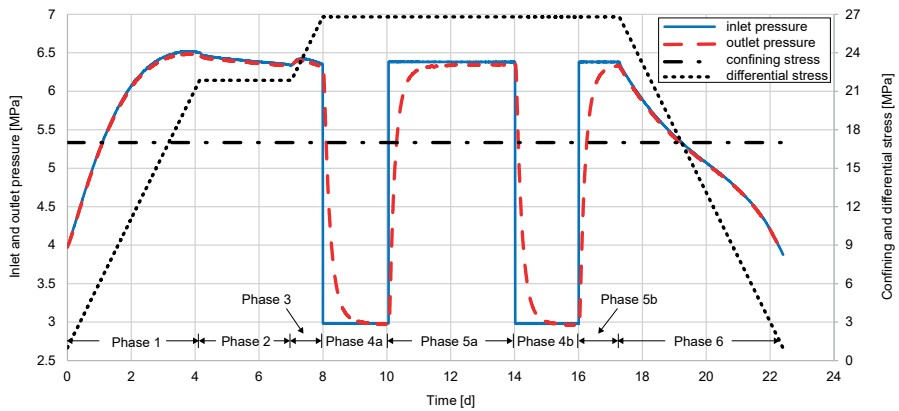

**Figure 4.** Proposed experimental procedure demonstrated on the Passwang Marl sample

Both methods rely on bringing the sample into the dilatant regime by changing the effective confining stress, which was shown to be critical for the crack damage stress (Wu et al., 2018). The effective confining stress $\sigma_3'$ is defined as

$$\sigma_3' = \sigma_3 - \alpha\mathbf{I}u \tag{4}$$

where $\sigma_3$ is the confining stress [MPa], $\alpha$ the Biot coefficient [-], $\mathbf{I}$ the identity matrix [-] and $u$ the pore pressure [MPa]. The effective confining stress is applied to the sides of the sample, while the differential stress $q$ is acting in axial direction and is

230 defined as:

$$q = \sigma_1 - \sigma_3 \tag{5}$$

where $\sigma_1$ is the normal stress acting in axial direction on the sample [MPa]. Changing the effective confining stress can be achieved by either changing the confining stress or the pore pressure within the sample. Both methods presented in this paper rely on changing the pore pressure to change the effective confining stress.

The new procedure, which combines the two methods, consists of several different phases in which the confining stress is kept constant throughout:

– Phase 1: In a CU test, the differential stress is increased until the (apparent) pore pressure maximum has been reached. Phase 1 is identical to the conventional way of determining the crack damage stress by the MPP method.

– Phase 2: All stresses are kept constant for a minimum of two days to detect any possible leakages (indicated by a pore

pressure decrease) and if necessary to determine a leakage correction term.

– Phase 3: The differential stress is increased again until the maximum pore pressure (after applying the leakage correction term) has been reached.

- Phase 4: In a CD test, the differential stress is kept constant, while the inlet pressure is decreased by typically several MPa. This phase lasts until the outlet pressure has stabilised.

- Phase 5: The inlet pressure is increased up to its value at the start of phase 4. Again, this phase lasts until the outlet pressure has stabilised. In this phase, the detection of the crack damage stress by the TAS method is possible.

- Phase 6: Under CU conditions, the differential stress is decreased to its value at the beginning of phase 1. This marks the end of one test cycle.

Depending on the response of the sample, phases 4 and 5 can be repeated several times in one test cycle, which in Fig. 4 is indicated by the annotations "a" and "b".

In the following, we show that the proposed new experimental procedure is the most effective way to gain accurate results and to overcome deficiencies of the conventional MPP method.

## 5.1  Maximum pore pressure method (MPP Method)

For the conventional MPP method, the differential stress in a CU test is increased until the maximum pore pressure is reached and the pore pressure starts to decrease (Phase 1 in Fig. 4). The point where pore pressure maximum occurs is then identified as the crack damage threshold and the acting axial stress as crack damage stress (Yoshinaka et al., 1997; Brandon et al., 2006). However, a closer look at this measurement technique revealed some weaknesses which can prevent an accurate detection of the crack damage stress. The most important one is a potential leakage which can develop in the course of an experiment even if the utmost care was taken to ensure a tight system at the start of the experiment. Such a leakage manifests itself as a drop in pore pressure, while all stresses are kept constant (Fig. 4, phase 2).

In the experiments used for this study, the leakage was so small (<0.1 ml/d) that it was impossible to detect its origin and to stop it. But even though the amount of fluid loss was tiny, the influence on the pore pressure was obvious and had to be taken into account for determining the crack damage stress. This was achieved by determining a leakage correction term using the pore pressure drop observed in phase 2. As in the initial phase of phase 2 the pore pressure drop is not only influenced by the leakage but also by a slight axial compaction of the sample due to time-dependent elasto-plastic processes, it is essential to wait until the decrease in pore pressure is relatively constant. In our experiments, two days were sufficient for phase 2 to establish a linear leakage correction term.

Another factor which negatively impacted our experiments were variations in room temperature which led to changes in pore pressure. This effect is also reported by other authors and usually causes pore pressure oscillations on a daily basis (Giger et al., 2018). Fig. 5 demonstrates how the changes in pore pressure due to room temperature variations can impact the accurate detection of the crack damage threshold.

## 5.2  True axial strain method (TAS method)

The crack damage stress depends on the effective confining stress (Wu et al., 2018) or the mean effective stress (Khaledi et al., 2023), respectively. If the confining stress is kept constant, there are two ways to change the effective confining stress

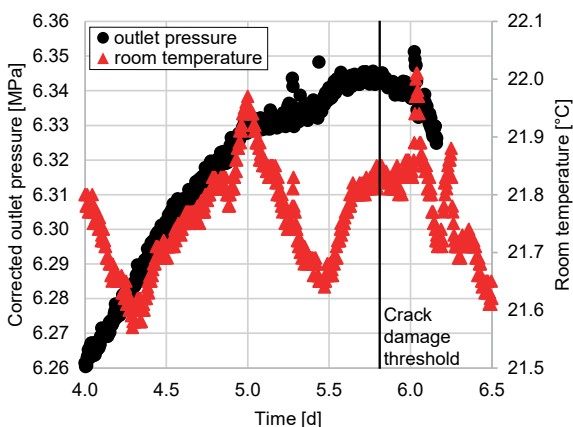

**Figure 5.** Crack damage threshold as detected by the MPP method (Passwang Marl) during phase 1

by manipulating the pore pressure: 1. the load is increased and as a result the pore pressure changes in a CU test; 2. the pore pressure is changed directly, while the load is kept constant in a CD test. While the MPP method uses the first possibility, the new TAS method relies on the second one.

For the new TAS method, the sample already has to be in the dilatant regime, which can be established by using the MPP method first (Fig. 4, phases 1 to 3). When the sample has crossed into the dilatant regime, all stresses (differential and confining stress) are kept constant and the pore pressure is first reduced by several MPa (Fig. 4, phase 4). This increases the effective confining stress and the sample crosses back into the non-dilatant regime. After the pore pressure has stabilised, it is increased back to its initial value (Fig. 4, phase 5), i.e. the effective confining stress is decreased and the sample again enters the dilatant regime. The crack damage threshold is then indicated by the minimum in true axial strain in phase 5. To be more precise, at the crack damage stress, the true axial strain stops decreasing and starts increasing (Fig. 6). For this study, positive true axial strains indicate compaction of the sample.

If the evolution of the true axial strain is analysed, a remarkable difference between the evolution at low and high differential stresses can be observed (Fig. 6a). The true axial strain showed a continuous decrease for increasing pore pressure within the sample for low differential stresses (below the dilatant regime). For high differential stresses however, where an increase in pore pressure resulted in a shift into the dilatant regime, the true axial strain exhibited a sudden increase after the initial decrease. The point where the true axial strain was minimal marked the crack damage threshold.

The change in true axial strain due to a pore pressure increase was smaller for high than for low differential stresses (Fig. 6a). However, the minimum of the curve could still be determined with a high accuracy. The crack damage threshold can also be clearly identified if the true axial strain is plotted against the duration of phase 5 (Fig. 6b).

Fig. 7 illustrates the different development of the main parameters of the MPP method (CU test) and the TAS method (CD test). Even though both methods can be used to detect the crack damage stress, a direct comparison is impossible due to the different stress paths and parameters involved for each method.

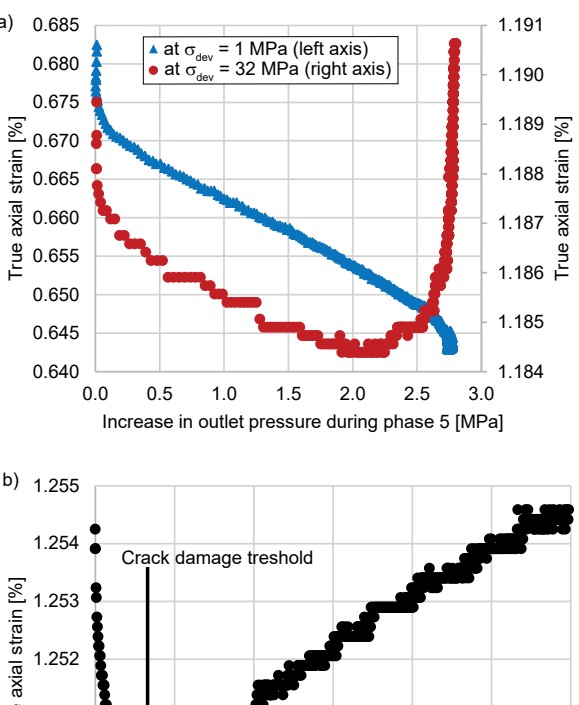

**Figure 6.** Evolution of true axial strain for Passwang Marl. a) Comparison for two differential stresses. Note the different ranges of the y-axes. b) crack damage threshold as seen in the true axial strain during phase 5

## 6  Discussion

### 6.1  Validity of TAS method

In order to prove the validity of the new method for detecting the crack damage stress, it has to be shown that the minimum in true axial strain during phase 5 indeed indicates the crack damage threshold, i.e. for each state of the system which fulfils the condition of the TAS method at this point

$$\frac{1}{\alpha} \cdot \frac{d\epsilon_{11}}{du}\bigg|_{\boldsymbol{\sigma}=const} = -\frac{d\epsilon_{11}}{d\sigma_1'}\bigg|_{d\sigma_3'=d\sigma_2'=d\sigma_1'} = 0 \tag{6}$$

also the condition for the crack damage stress according to the MPP method (phases 1 and 3)

$$\frac{d\epsilon_v}{d\sigma_1'}\bigg|_{\sigma_2=\sigma_3=const} = 0 \tag{7}$$

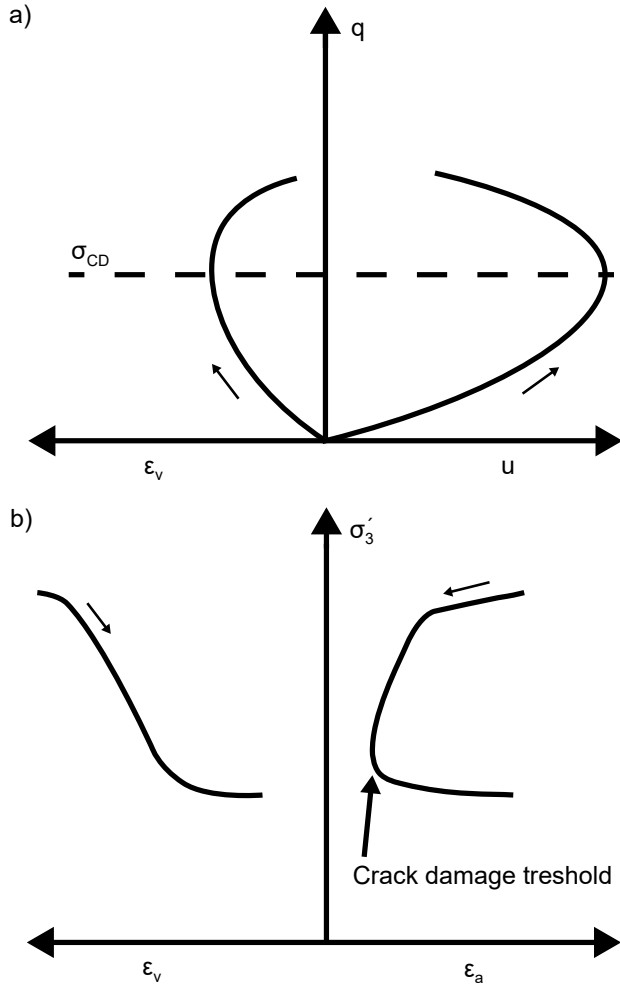

**Figure 7.** Sketch of the evolution of the main parameters during the experiment a) MPP method. b) TAS method. Small arrows indicate the direction of the process.

is fulfilled. Since at the point of the pore pressure maximum $du = 0$ holds, Eq. 7 becomes equivalent to:

$$\frac{d\epsilon_v}{d\sigma_1'}\bigg|_{\sigma_2'=\sigma_3'=const} = 0 \tag{8}$$

In general, the relationship between the differential change of the effective stress ($d\boldsymbol{\sigma}'$) and the differential change of the strain ($d\boldsymbol{\epsilon}$) is described by the constitutive equation

$$d\boldsymbol{\epsilon} = \boldsymbol{S} \cdot d\boldsymbol{\sigma}' \tag{9}$$

where $\boldsymbol{S}$ is the compliance matrix of the material (a 4th order tensor), which is identical to the inverse of the stiffness tensor $\boldsymbol{C}$ (i.e. $\boldsymbol{S} = \boldsymbol{C}^{-1}$).

Using the notation of Voigt, Eq. 9 gets the form

$$\begin{bmatrix} d\epsilon_{11} \\ d\epsilon_{22} \\ d\epsilon_{33} \\ 2d\epsilon_{23} \\ 2d\epsilon_{13} \\ 2d\epsilon_{12} \end{bmatrix} = \begin{bmatrix} S_{11} & S_{12} & S_{13} & S_{14} & S_{15} & S_{16} \\ S_{21} & S_{22} & S_{23} & S_{24} & S_{25} & S_{26} \\ S_{31} & S_{32} & S_{33} & S_{34} & S_{35} & S_{36} \\ S_{41} & S_{42} & S_{43} & S_{44} & S_{45} & S_{46} \\ S_{51} & S_{52} & S_{53} & S_{54} & S_{55} & S_{56} \\ S_{61} & S_{62} & S_{63} & S_{64} & S_{65} & S_{66} \end{bmatrix} \begin{bmatrix} d\sigma_{11}' \\ d\sigma_{22}' \\ d\sigma_{33}' \\ d\sigma_{23}' \\ d\sigma_{13}' \\ d\sigma_{12}' \end{bmatrix} \tag{10}$$

It has to be kept in mind that the Voigt notation of the compliance matrix is symmetrical in all coordinate systems and for
arbitrary anisotropies of the material, i.e.

$$S_{ij} = S_{ji} \tag{11}$$

is always true. In its principal coordinate system, the stress change is

$$d\boldsymbol{\sigma}' = \begin{bmatrix} d\sigma_1' & 0 & 0 \\ 0 & d\sigma_2' & 0 \\ 0 & 0 & d\sigma_3' \end{bmatrix} \tag{12}$$

In this coordinate system, from Eqs. 10 and 12 follows for the diagonal elements of the strain tensor

$$d\epsilon_{ii} = \sum_{j=1}^{3} S_{ij} \cdot d\sigma_j' \tag{13}$$

which gives for the volumetric strain

$$d\epsilon_v = \sum_{i=1}^{3} d\epsilon_{ii} = \sum_{i=1}^{3}\sum_{j=1}^{3} S_{ij} \cdot d\sigma_j' \tag{14}$$

For the conditions of the loading path of the TAS method (i.e. $d\sigma_3' = d\sigma_2' = d\sigma_1'$), Eq. 13 yields

$$d\epsilon_{11}\big|_{d\sigma_3'=d\sigma_2'=d\sigma_1'} = d\sigma_1' \cdot \sum_{j=1}^{3} S_{1j} \tag{15}$$

$$\Rightarrow \frac{d\epsilon_{11}}{d\sigma_1'}\bigg|_{d\sigma_3'=d\sigma_2'=d\sigma_1'} = \sum_{j=1}^{3} S_{1j} = S_{11} + S_{12} + S_{13} \tag{16}$$

For the condition of the loading path of the crack damage stress (i.e. $\sigma_2' = \sigma_3' = const \Leftrightarrow d\sigma_2' = d\sigma_3' = 0$), Eq. 14 yields

$$d\epsilon_v\big|_{\sigma_2'=\sigma_3'=const} = d\sigma_1' \cdot \sum_{i=1}^{3} S_{i1} \tag{17}$$

$$\Rightarrow \frac{d\epsilon_v}{d\sigma_1'}\bigg|_{\sigma_2'=\sigma_3'=const} = \sum_{i=1}^{3} S_{i1} = S_{11} + S_{21} + S_{31} \tag{18}$$

Due to the symmetry of the compliance matrix (Eq. 11), Eq. 18 is equivalent to

$$\frac{d\epsilon_v}{d\sigma_1'}\bigg|_{\sigma_2'=\sigma_3'=const} = \sum_{i=1}^{3} S_{i1} = S_{11} + S_{12} + S_{13} \tag{19}$$

Using Eqs. 15 and 19, it follows that

$$\frac{d\epsilon_{11}}{d\sigma_1'}\bigg|_{d\sigma_3'=d\sigma_2'=d\sigma_1'} = \frac{d\epsilon_v}{d\sigma_1'}\bigg|_{\sigma_2'=\sigma_3'=const} \tag{20}$$

This shows that Eq. 6 and 8 are equivalent.

It has to be noted that in this argumentation both conditions refer to the same system state. Thus, this argumentation is also valid if the compliance tensor depends on the stress state and perhaps also the stress history.

## 6.2 True axial strain versus true volumetric strain

The physical derivation of the behaviour of the true axial strain at the crack damage stress using the TAS method is confirmed by experimental observations. Fig. 6a shows that during phases of pore pressure increases the true axial strain evolves differently for samples under high and low differential stresses. Under high differential stress, the true axial strain changes from decrease to increase (i.e. compaction) for increasing pore pressure.

The results of the experiment with fitted extensiometer chains (Fig. 8a) demonstrate that the radial strain continues to decrease even while the true axial strain begins to increase. Equivalent volumetric strains are calculated using the true axial strain as well as the radial strain measured by only one extensiometer chain as input data. Most of the radial strain occurs in the upper part of the Bunter Sandstone sample (illustrated by the large decay in equivalent volumetric strain), indicating that dilatancy is not uniformly distributed over the sample's length. The true volumetric strain shows a continuous decrease even though the true axial strain decreases at the beginning of phase 5 followed by an increase (Fig. 8b). Thus, the sample proceeds to expand in volume despite the fact that compaction occurs in axial direction.

## 6.3 Multiple measurements of crack damage stress

Results from both methods are displayed for comparison in Fig. 9 for several test cycles of Passwang Marl. Values which are derived from the same test cycle are connected by dashed lines. The onsets of dilatancy derived by both methods cannot be identical as the sample undergoes plastic deformation between the measurements of a single test cycle. However, the very similar gradients of the connecting lines indicate that both methods yield consistent results even if a certain amount of plastic

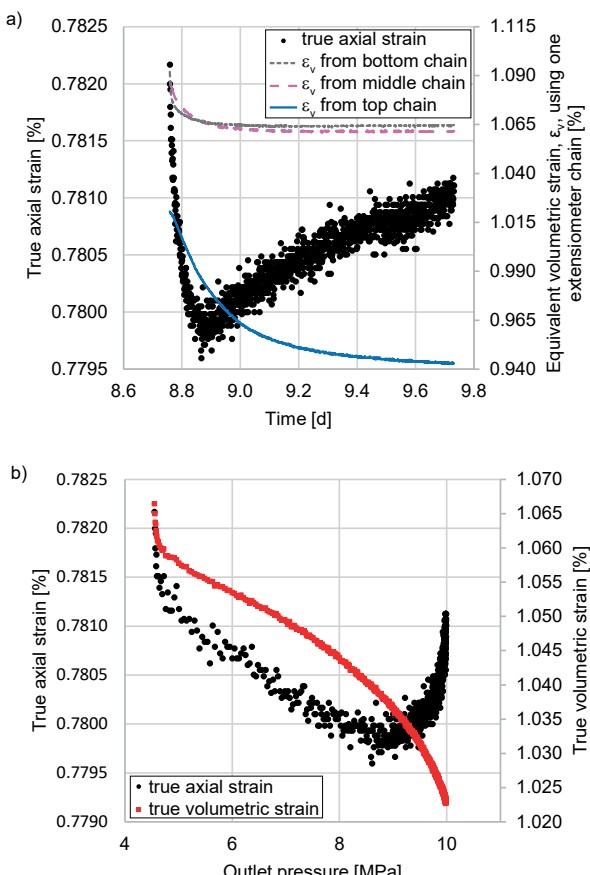

**Figure 8.** Evolution of true axial strain for Bunter Sandstone. a) crack damage stress as seen in the true axial strain during phase 5. b) Comparison of true axial strain and true volumetric strain evolution. Note the different ranges of the y-axes

deformation occurred between the measurements. Therefore, multiple testing of the same sample with respect to the crack

damage stress is possible and produces reliable results. As a consequence, it is advantageous to use measurement techniques - such as the proposed TAS method - which allow multiple measurements in a comparatively short period of time. The stresses depicted for the crack damage stress are the differential stresses ($\sigma_1 - \sigma_3$), which occur at pore pressure maximum (MPP method) or true axial strain minimum (TAS method).

   In Fig. 10 the results of both methods are displayed for the Bunter Sandstone sample. The results for the true axial strain

(Fig. 10a) as well as for the true volumetric strain (Fig. 10b) are shown as for this sample also true volumetric strains were available. Here, the dashed lines connect the results derived by the MPP method (Fig. 4, phase 1) and the first results from the TAS method (Fig. 4, phase 5a). Subsequent results measured by the TAS method in the same cycle but in different phases (phases 5b and following) are displayed but not connected by dashed lines.

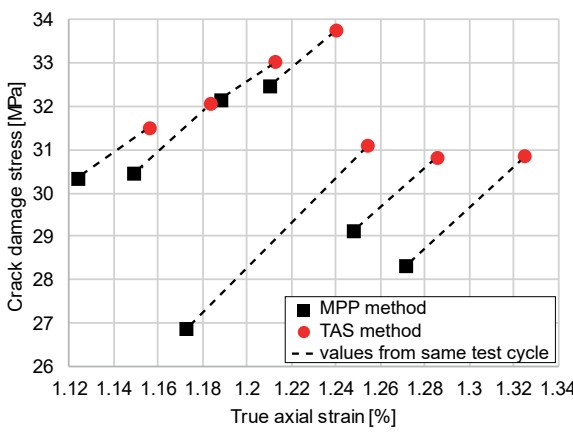

**Figure 9.** Comparison of crack damage stresses detected with both methods for Passwang Marl

Further proof that both methods yield reliable results and that fast, multiple measurements of the crack damage stress are necessary is shown in Fig. 11 and in a movie in the supplementary material. Both demonstrate that the crack damage stress does not depend on one parameter but on two - the effective confining stress and the true volumetric strain. If both parameters are considered in such a bilinear fit taking into account the results of both methods, the coefficient of correlation is 0.990 (compared to 0.626 for Fig. 10b), demonstrating impressively that the results of both methods match. However, this bilinear behaviour of the crack damage stress necessitates multiple measurements which cover a significant section of the state space to determine the dilatant behaviour of the sample in a robust manner.

### 6.4 Advantages of the TAS method

The TAS method proposed in this study offers the possibility to achieve more accurate measurements of the crack damage stress than before by not only combining two measurements techniques which are completely independent but also by offering the chance to quantify and correct some problems affecting the conventional MPP method. Using two different machine setups to demonstrate the TAS method also shows that it is applicable under varying measurement conditions as the detection of the crack damage stress was possible in both experiments. Moreover, the application of the new TAS method yields a maximum of information in a minimum of time. Below, we will discuss these points in more detail.

#### 6.4.1 More accurate and faster measurements

The true axial strain is unaffected by changes in room temperature as the relevant components and measuring equipment are located inside the temperature regulated inner cell of the triaxial testing apparatus. Thus, the signal which is used for detecting the crack damage stress is free of influences caused by outside sources. Consequently, the crack damage stress can be detected with much greater accuracy than with the MPP method.

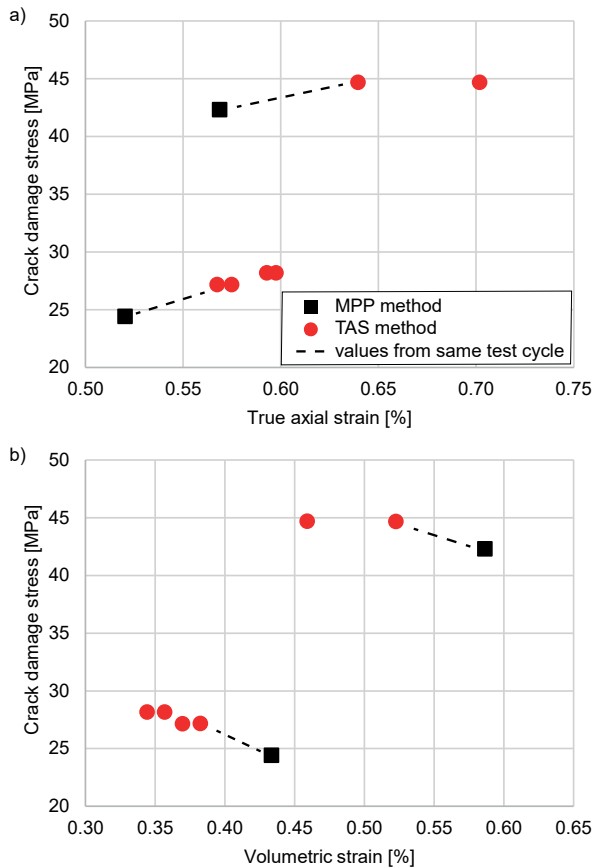

**Figure 10.** Comparison of crack damage stresses detected with both methods for Bunter Sandstone; a) results with respect to the true axial strain, b) results with respect to the volumetric strain

Moreover, the detection of the crack damage stress by the TAS method is free of any leakage influence. Any leakage can be compensated by the pressure pump and thus does not influence the results as the pore pressure within the sample is actively regulated during phase 5, in which the detection takes place. Hence, a critical factor which can significantly complicate detecting the crack damage stress by the MPP method is not relevant for the new method.

The MPP method is not only hampered by external influences which impede the precise determination of the crack damage stress but also by the time lag occurring between the crack damage stress in the centre of the sample, where the differential stress due to friction at the front ends is maximal, and its detection at the front ends of the sample. For materials such as the Bunter Sandstone, this time lag in detection may be small, but for materials with lower permeability such as the Passwang Marl used in this study, it can be significant.

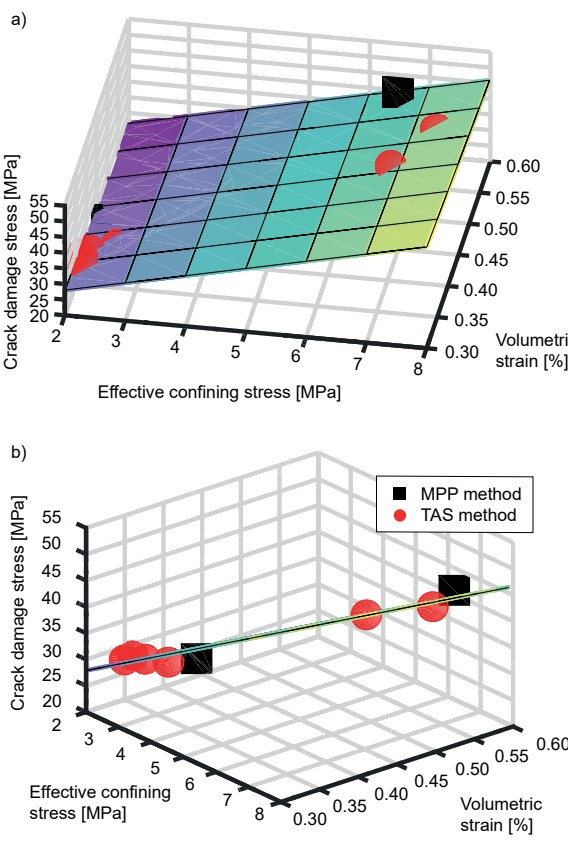

**Figure 11.** Bilinear fits for the dependence of the crack damage stress on effective confining stress and volumetric strain for Bunter Sandstone under two different viewing angles

The true axial strain does not experience such a time lag and therefore an immediate detection of the crack damage stress is possible. Thus, the TAS methods allows for a more accurate identification of the crack damage stress as the uncertainty of the results is much smaller with e.g. 0.02 MPa compared to 1.2 MPa for the MPP method.

An additional benefit is that as the true axial strain is measured over the entire sample length; thus, the crack damage stress detected by its minimum gives a value which integrates over the entire sample. It therefore avoids the discrepancies between inlet and outlet pressure signal which may occur for certain samples e.g. the Bunter Sandstone used in this study.

### 6.4.2   Gain in information

The TAS method not only enables the determination of the crack damage stress but also of pressure diffusion coefficients,
which can be used to calculate permeabilities as they are identical to coefficients of consolation (the difference being the method by which they are determined). (Wang, 1993; Robinson, 1998; Abuel-Naga and Pender, 2012; Di Francesco, 2013). In

contrast to the crack damage stress, where only phase 5 yields results, pressure diffusion coefficients can be calculated from the outlet pressure changes in phases 4 and 5. Thus, a full test cycle can provide a plethora of data. The test cycle depicted in Fig. 4 provided four validated values for the crack damage stress (phases 1, 3, 5a and 5b) and four pressure diffusion coefficients (phases 4a to 5b), thereby maximising the information output for this type of experiment.

### 6.4.3 Independent verification

The TAS method can provide an independent verification of the results of the MPP method as it is not subjected to unwanted influences on the results such as room temperature changes or leakage. The quality of the results of the MPP method depends on the correct application of correction terms and is thus more susceptible to errors. However, if the results of both methods match as well as in Fig. 9, it can be concluded that the results obtained by the MPP method are reliable and that the applied correction terms are correct. This is especially important for experiments with runtimes in the range of months to years, in which leakages may develop over time. These experiments do not need to be stopped because of the leakage but can be continued and the quality of their results continually verified via the TAS method.

## 7 Conclusions

The proposed new experimental procedure which combines two techniques - the well known MPP method (for CU tests) and the newly developed TAS method (for CD tests) - enables a faster and more accurate detection of the crack damage stress in fully saturated samples than before. Both factors are of high relevance for the determination of a constitutive equation for the crack damage stress.

Especially with the new TAS method presented in this study, a more precise determination of the crack damage stress is possible. For the TAS method, a pressure diffusion test is performed where the pore pressure is increased at high constant axial loads. The crack damage stress is reached when the true axial strain changes its trend from expansion to compaction during the pore pressure increase. Potentially occurring leakages, which e.g. can hamper the detection of the crack damage stress with the MPP method, cannot disturb the measurement as for the TAS method the pore pressure is actively regulated. Thus, the TAS method allows for a more robust determination of the crack damage stress than the MPP method.

Additionally, this new technique not only provides information on the crack damage stress but also allows the determination of pore pressure diffusion coefficients and thus permeabilities at the same time.

The proposed new experimental procedure is applicable to all types of fully saturated, permeable rocks as has been successfully demonstrated by applying it to hydraulically tight Passwang Marl as well as permeable Bunter Sandstone.

*Data availability.* The data that support the findings of this study are openly available in Zenodo at https://zenodo.org for the Passwang Marl, Z-orientation (https://doi.org/10.5281/zenodo.4063577) (Schumacher and Gräsle, 2020b) and the Bunter Sandstone (https://doi.org/10.5281/zenodo.40638 (Schumacher and Gräsle, 2020a).

*Video supplement.* An additional supporting video can be found on Zenodo (https://doi.org/10.5281/zenodo.14412332).

*Author contributions.* Sandra Schumacher: Conceptualization, Methodology, Formal analysis, Investigation, Writing - original draft, Visualization. Werner Gräsle: Conceptualization, Formal analysis, Writing - review and editing.

*Competing interests.* The authors declare no potential conflict of interests.

*Acknowledgements.* The authors thank Uwe Brockmann and Sebastian Klösters for operating and maintaining the triaxial testing apparatus. We also thank Herbert Röhm from the State Authority for Mining, Energy and Geology, Lower Saxony, for providing the Bunter Sandstone sample and Kristian Ufer (BGR) for its mineralogical analysis.

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
