# Peer review of "Crack damage stress in fully saturated rocks: A new detection procedure"

_EGUsphere, 2024_

## Author Response (AR1)

Reply to the comments

Reply to Giacomo Medici

Thank you for your comments on our paper, which helped us to improve it. We hope that our changes to the manuscript will find your approval.

Specific comments

Lines 11-12. "Understanding the hydro-mechanical behaviour of rocks...construction". Statement not backed up by references. Please, insert the following review on energy extraction and nuclear waste repositories:

- Medici G., Ling F., Shang J. 2023. Review of discrete fracture network characterization for geothermal energy extraction. Frontiers in Earth Science, 11, doi.org/10.3389/feart.2023.1328397

 - David, C., & Le Ravalec-Dupin, M. (2007). Rock physics and geomechanics in the study of reservoirs and repositories. Geological Society, London, Special Publications, 284(1), 1-14.

We have included the required references.

Lines 11-12. The hydro-mechanical behaviour of rocks is also relevant in geothermal energy. Please, specify this point (see reference above).

We have done that.

Lines 12-18. Several other statements are not back up by references.

We have included several references to back up the statements.

Line 55. Clearly disclose the aim of your research and the 3 to 4 specific objectives by using numbers (e.g., i, ii and iii) at the end of your introduction.

We have rewritten this part to clarify our objectives.

Lines 57-onwards. Provide detail on the sedimentological characteristics of the two samples. For example the sample of the Bunter Sandstone is fluvial or aeolian? Age?

We have now included information on the sedimentology and age of the samples.

Lines 57-onwards. Have you got information on the petrophysical (e.g., porosity, and hydraulic conductivity) and mineralogical components of those samples from previous studies or theses? You mention the concept of permeability in your manuscript.

We have now included information on the petrology and mineralogy of the samples.

Lines 359-366. Expand the conclusions which are too short.

We have expanded the conclusion to include more details on the TAS method.

Figures and tables

Figures 1, 3, 4, 5. Time units unclear and undefined. They also change (h and d) in the figures. Please, check if you have provided the relevant information in the main body of your manuscript.

We have replaced one figure so that now all figures use the unit [d]. Moreover, we have included information regarding the time units in the manuscript.

Figures 7-9. Make the figure larger, there is room for doing that.

As we had already chosen the maximum width, which this journal allows for two column figures, we have changed the format of the figures from two-column landscape to one-column portrait for enlargement. The subfigures are now displayed atop each other instead of next to each other.

Figure 10. Make the figure larger and provide more detail in the caption ("Bilinear fits of....").

We have enlarged the figure and included more information in the caption.

Reply to Thomas Poulet,

Thank you very much for your thorough review of our paper. We appreciate your concerns and addressed them accordingly.

Specific comments:

1. The writing style would benefit from some editing. While the journal might provide some help for the phrasing, I do recommend the author modify some aspects themselves, as there are aspects which make the document look a bit too much like a report rather than a scientific paper.

    1. The abstract must be rewritten with a revised structure to present the information in a more conventional order (see https://www.nature.com/documents/nature-summary-paragraph.pdf).

        We have rewritten the abstract so that it conforms to a conventional order.

    2. The pitch of the introduction should be changed to highlight more clearly the problem tackled and the hypothesis tested, instead of remaining descriptive and mentioning in the last paragraph that several problems were identified (without saying which ones). L.48 is the justification of the whole paper, so it must be clear and explicit. The current writing makes it sound too much like a report with serendipitous findings, which doesn't really do justice to the good work.

We have changed the pitch of the introduction to more clearly state the problem and our hypothesis.

3. Paragraphs should be broken by meaning. (e.g., in the introduction, the last sentence of the first paragraph and the first sentence of the second paragraph shouldn't be separated. Move the last sentence of the first paragraph to the second paragraph.)

   We have corrected this paragraph and all other broken paragraphs.

4. The current writing style tends to favour starting sentences with subordinates, which does not necessarily facilitate the reading.

   We have changed the sentence structure in many cases in order to significantly reduce the number of sentences which start with a subordinate.

5. I would recommend using one of the main free solutions to check and potentially improve the writing (e.g., typos l.243 "proof" and l.279 "has to [be] noted).

   The typos have been corrected and an additional check has been performed.

2. Keeping the word "permeable" in the title doesn't really make sense to me without a discussion on the topic of permeability or porosity threshold below which the technique is not expected to perform well any longer. (Note that the word "permeable" is probably not prescriptive enough in any case, as rocks with $10^{-20}$ m$^2$ permeability are still "permeable"...).

   You are right. The Passwang Marl we have used for our study would normally be considered to be impermeable given its very low permeability in the range of $10^{-20}$ to $10^{-19}$ m². We have thus dropped the word "permeable" in the title as "fully saturated" already implies a certain permeability, however small it may be.

3. The description of the dilatancy and compaction regimes is not particularly good (l.16-18). Dilation must be explained properly. Note that a dilatant or compactant response of the rock in plasticity depends on the loading path. L.19: the authors don't explain the bulk volume evolution, which decreases and then increases, marking the "crack damage" threshold. It should be linked to the micro-scale evolution of micro-cracks. The "crack damage" can be defined as the onset of crack coalescence to form macroscopic fractures, see e.g. Cai et al 2004 https://doi.org/10.1016/j.ijrmms.2004.02.001). The authors might consider reordering the micro- and macro-scale descriptions, as described from l.40 onwards. Starting with a macroscopic view without schematic or microscopic description doesn't make sense for readers not already familiar with all the concepts (, in which case they don't need to read this at all).

   We have rewritten the whole part and included a more detailed microscopic description of the crack development under load. Moreover, we rearranged the paragraphs so that now the microscopic explanation comes before the macroscopic observations.

4. L.25, define the instantaneous Poisson value .

   We have included the definition of the instantaneous Poisson ration.

5. L.27 is assuming that the reader is familiar with axial stress-volumetric strain curves. All mentions are most welcome but would be more easily understood with some schematic

descriptions. => A figure is needed to depict the general concepts described in the text of the introduction (e.g. see Cai et al 2004 again).

We have included a new figure showing the stress-strain development as well as pore pressure evolution and volumetric strain evolution. The curves shown there are partially based on our measurements of the Bunter Sandstone; partially as our own measurements did not include sample failure. However, the data we have are more than sufficient to show the concept of the different stages of crack development.

6. L.58: mention why two apparatuses were needed for two separate samples. At first sight, that doesn't provide confidence in the generalisation claim about the method from the abstract. ;-)

We have included an explanation. Actually, we could also have included data from a third sample (Opalinus Clay), which was tested with an identical apparatus than that for the Passwang Marl. We refrained from this as it would not have substantiated our claim any further just added more data.

7. L.67: can you comment on not using circumferential measurements? Please justify why two different measurement methods were used for the two experiments (pre-empting the Discussion and presenting this as a hypothesis to test rather than just a coincidence that the two apparatuses were set up differently).

We have included this information.

8. L.97: any reference for readers interested in more precision about that borehole?

We have included a reference.

9. L.170: not clear at this stage what phases 1-3, 4, 5 denote. (Add "detailed below")

We have included the necessary information.

Reply to reviewer 2

Thank you for your comments, which we have addressed to improve the paper.

1. L.12-15, at the beginning of the introduction, provide a detailed explanation of the relationship between crack damage stress ($\sigma_{CD}$) and dilatancy, emphasizing their importance in rock mechanics.

We have rewritten this paragraph and also included an additional figure in order to show the relationship between crack damage stress and dilatancy.

2. The research objectives and hypotheses are not explicitly stated. Conclude the introduction with a clear list of objectives.

We have now included a list of objectives (points (i) to (iii)) close to the end of the introduction.

3. L.68-70, the description of the experimental setup, particularly the hydraulic system and porous discs of sintered metal, is too vague. Please provide more detailed information about the functioning and roles of each component in the experiment.

We have included more details about the hydraulic system and the porous discs in order to clarify their roles.

4. The method for detecting crack damage stress ($\sigma_{CD}$) is not clearly explained.

Unfortunately, this remark is too unspecific for us to act upon. The whole paper deals with the explanation of how to detect the crack damage stress and according to this reviewer, the paper is convincing. Therefore, it is difficult to know where and how we have to improve the paper.

5. The rationale for using different setups for two samples is not adequately discussed. Explain why different setups are necessary and discuss how this design affects the generalizability of the results.

We have included a brief explanation why two different setups have been used. Different setups were not necessary and in this case just a result of the available triaxial machines. The Bunter Sandstone experiment was deliberately set up to provide the maximum information to verify the new method, while the Passwang Marl experiment was an already running experiment with data to support our claim. By chance using two different setups shoes that the kind of setup does not change the outcome of the results as can be expected as the theoretical derivation of the new method (section 6.1) shows it to be universally applicable independent of the machine setup.

6. Include a brief definition for "differential stress" and "effective confining stress" for clarity.

We have included a definition for both parameters in the text.